# Learning Compact Reward for Image Captioning

## Abstract

Adversarial learning has shown its advances in generating natural and diverse descriptions in image captioning. However, the learned reward of existing adversarial methods is vague and ill-defined due to the reward ambiguity problem. In this paper, we propose a refined Adversarial Inverse Reinforcement Learning (rAIRL) method to handle the reward ambiguity problem by disentangling reward for each word in a sentence, as well as achieve stable adversarial training by refining the loss function to shift the stationary point towards Nash equilibrium. In addition, we introduce a conditional term in the loss function to mitigate mode collapse and to increase the diversity of the generated descriptions. Our experiments on MS COCO show that our method can learn compact reward for image captioning.

## 1 Introduction

Image captioning is a task of generating descriptions of a given image in natural language. In a general encoder-decoder structure (Vinyals et al., 2015), image features are encoded in a CNN and decoded into a caption in a word by word manner. Based on the loss function, standard approaches to the problem could be divided into three categories: MLE (Maximum Likelihood Estimation), RL (Reinforcement Learning) and GAN (Generative Adversarial Network).

Early proposed methods were based on MLE function and made improvements by designing specific model structure (Xu et al., 2015). MLE adopts the cross-entropy loss and learns a one-hot distribution for each word in the sentence. By maximizing the probability of the ground truth word whilst suppressing other reasonable vocabularies, the probability distribution learned by MLE tends to be *sparse* and the generated captions have limited diversity (Dai et al., 2017). On the other hand, RL has advantages in boosting the model performance by optimizing the handcrafted metrics (Rennie et al., 2017; Liu et al., 2017; Chen et al., 2019). However, due to the reward hacking problem, RL maximizes the reward in an unintended way and fails to produce human-like descriptions (Li et al., 2019a). Considering naturalness and diversity of the generated captions, GAN has raised attention in image captioning for its capability of producing descriptions that are indistinguishable from human-written ones (Dai et al., 2017; Shetty et al., 2017; Chen et al., 2019; Dognin et al., 2019).

In image captioning, the generator of GAN learns true data distribution by maximizing the reward function learned from a discriminator, and the discriminator distinguishes the generated sample from the true data. The adversarial training converges to an equilibrium point (i.e., Nash equilibrium) at which both the generator and discriminator cannot improve (Goodfellow et al., 2014). As shown in Figure 1, the learned distribution of GAN is closer to the ground truth distribution than that of other methods (i.e., MLE and RL) on different splits. However, previous work of adversarial networks in image captioning gives one reward function $D$ for a complete sentence consisting of $n$ words. This strategy causes the reward ambiguity problem (Ng et al., 1999) since which word(s) causes the reward to increase or decrease is not accounted for, and thus there are many optimal policies that determine the sentence can explain one reward. An example is that each image in MS COCO has five ground truth captions. Although these captions may vary in formats, each caption has the same reward value in GAN. From the perspective on the system level, learning sentence-level reward from different image-caption pairs is analogous to learning reward of a trajectory from different system dynamics, which makes the discriminator unable to distinguish the true reward functions from those shaped by the environment dynamics (Fu et al., 2018).

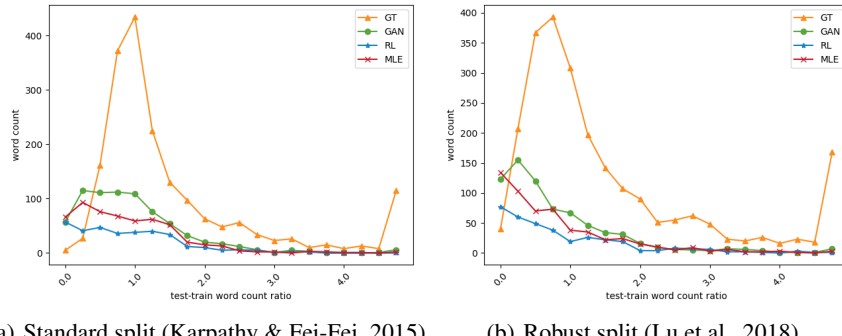

(a) Standard split (Karpathy & Fei-Fei, 2015)      (b) Robust split (Lu et al., 2018)

Figure 1: Comparison of word count ratios (Shetty et al., 2017) on two splits of MS COCO. $x$ axis is the {test frequency}/{train frequency} of a word and $y$ axis is the word count of the corresponding ratio. GT represents ground truth distribution.

Facing the challenge, we adopt AIRL (Fu et al., 2018) to solve the reward ambiguity problem by disentangling reward for each action (i.e., word in a sentence) from different image-caption pairs and learning a compact reward function. *compact* means words with similar semantics, such as *children* and *kids*, correspond to close reward values. Driven by the compact reward function of the discriminator, the generator learns the optimal policy and thus produces qualitative descriptions. However, there are still two major problems to address: 1) AIRL is difficult to converge to Nash equilibrium using policy gradient (See Section 4.2 for details); 2) AIRL is designed without mode control, and thus the outputs have limited diversity, which is a commonly encountered issue called mode collapse (Mirza & Osindero, 2014).

In this paper, we propose a refined AIRL method to learn a compact reward function for each word, as well as achieve stable adversarial training by refining the loss function to shift the stationary point towards Nash equilibrium. The refined method makes it possible to reach the equilibrium point for a non-concave model function of the generator. In addition, a conditional term is introduced in the loss function to mitigate mode collapse and to increase the diversity of the generated descriptions. Both the caption evaluator (i.e., discriminator) (Cui et al., 2018; Sharif et al., 2018) and the generator are cast into this unified framework, where the discriminator evaluates captions using a learned compact reward function, and the generator produces qualitative image descriptions. We demonstrate the effectiveness of our method in the experiments.

## 2 RELATED WORK

**Image Captioning.** The development of image captioning can be summarized into two directions: model structure design (Lu et al., 2017; Yao et al., 2018) and loss function construction (Rennie et al., 2017; Ren et al., 2017). In the methods based on model structure design, attention mechanism and the fusion of visual and semantic information are the key focus. Lu et al. (2017; 2018) proposed a sentinel gate to learn adaptive attention between visual content and non-visual text. Yao et al. (2018) explored the role of visual relationship in image captioning. On the other hand, methods based on loss function construction focus on optimization of the loss function. Rennie et al. (2017) optimized on non-differentiable evaluation metric using policy gradient, and improved scores of these metrics on various models. Ren et al. (2017) designed an embedding reward under actor-critic reinforcement learning. Similarly, we address the construction of loss functions, and thus our algorithm can be built on existing model structures. See Appendix F for a short discussion about different loss functions.

**Adversarial Methods for Image Captioning.** Adversarial methods are known for producing plausible samples by training the generator and the discriminator in an adversarial manner (Goodfellow et al., 2014). In image captioning, the discriminator is formed as a binary classifier that distinguishes the generated sentence from the ground truth, while the generator produce captions that can fool the discriminator. Conditional GAN was proposed in (Dai et al., 2017) to improve the naturalness and diversity of generated captions. CNN and RNN based discriminators were introduced in (Chen et al., 2019). However, existing methods estimate a reward function for the complete sentence consisting of $n$ words, where multiple optimal policies that determine the sentences can correspond to one reward

(Ng et al., 1999). Thus the learned reward is ambiguous and ill-defined. We solve this problem by recovering a compact reward function for each word in the sentence under a refined AIRL framework. Although AIRL has been utilized to solve problems in other fields (Wang et al., 2018; Li et al., 2019b; Shi et al., 2018), we are the first to make algorithmic improvements to AIRL such that Nash equilibrium can be reached even for a non-concave model function of the generator, and that diversity of the outputs can be increased.

## 3 ADVERSARIAL INVERSE REINFORCEMENT LEARNING

Due to the high variance estimate of a full sentence and the reward ambiguity problem, instead of learning reward for a complete sentence, we could learn reward distribution $p_\theta(w_t, s_t)$ for each word-state pair $(w_t, s_t)$ so that the true reward can be recovered at optimality (Fu et al., 2018). In the following, we introduce how AIRL disentangles reward for each word-state pair $(w_t, s_t)$.

AIRL is an adversarial reward learning algorithm based on Maximum-Entropy-IRL. Finn et al. (2016) first proved that Maximum-Entropy-IRL is mathematically equivalent to GAN under a special form of the discriminator:

$$D_\theta(w_t, s_t) = \frac{p_\theta(w_t, s_t)}{p_\theta(w_t, s_t) + \pi(w_t, s_t)} \tag{1}$$

$$p_\theta(w_t, s_t) = \exp\{f_\theta(w_t, s_t)\} \tag{2}$$

where $p_\theta(w_t, s_t)$ is the actual probability distribution estimated by the discriminator. $\pi(w_t, s_t)$ is the policy distribution produced by the generator, which is also called vocabulary distribution under the context of image captioning.

The goal is to estimate $p_\theta(w_t, s_t)$ that approximates the true data distribution $p_{\text{true}}(w_t, s_t)$, as well as to learn an optimal vocabulary distribution $\pi$ that maximizes the reward. Subsequently, considering reward ambiguity problem, Fu et al. (2018) further extended the theory to AIRL by adding a reward shaping term $h_\varphi$ into $f_\theta(w_t, s_t)$, and thus disentangled reward from different system dynamics (i.e., image-caption pairs):

$$f_{\theta,\varphi}(w_t, s_t) = g_\theta(w_t, s_t; s_{t+1}) + \gamma h_\varphi(s_{t+1}) - h_\varphi(s_t) \tag{3}$$

where $g_\theta$ denotes the reward approximator that recovers the true reward up to a constant, and $h_\varphi$ is the reward shaping term that preserves the optimal $\pi$. $\gamma$ is a constant in range $(0, 1]$.

In the context of divergence minimization, the adversarial process can be represented as a min-max game (Mescheder & Geiger, 2017):

$$\min_\pi \max_{\theta,\varphi} \mathbb{E}_{w_t^{\text{true}} \sim p_{\text{true}}}[\log\left(D_{\theta,\varphi}(w_t^{\text{true}}, s_t^{\text{true}})\right)] + \mathbb{E}_{w_t \sim \pi}[\log\left(1 - D_{\theta,\varphi}(w_t, s_t)\right)] \tag{4}$$

where $p_{\text{true}}$ is the true data distribution and $\pi$ is the vocabulary distribution learned by the generator. $(w_t^{\text{true}}, s_t^{\text{true}})$ is the word-state pair of the true data.

Despite of the capability of AIRL in disentangling reward for each word, it is difficult for the above AIRL algorithm to converge to Nash equilibrium and to produce diverse outputs through adversarial training (See Section 4.2 for details). These issues can result in a non-optimal solution and lack of diversity of the generated descriptions. Aiming to learn the optimal compact reward as well as diverse captions, we refine the loss function to shift the stationary point towards Nash equilibrium and to mitigate mode collapse in the two-player game.

## 4 LEARNING COMPACT REWARD FOR IMAGE CAPTIONING

To address the problems discussed above, we refine the loss function to: *1)* find a compact reward function that is optimal; *2)* increase diversity of the generated captions. In particular, a *constant term* is used to solve *1)* by shifting the stationary point to Nash equilibrium, and a *conditional term* is introduced to solve *2)* by utilizing mode control techniques. Our algorithm is detailed in Algorithm 1, where $n$ is the sentence length and $N$ denotes number of iterations.

In the following notations, $\theta$ and $\varphi$ are the parameters of the discriminator, and $\psi$ represents the parameter of the generator. $w_t$ and $s_t$ denote the $t_{th}$ word and its corresponding hidden state vector, respectively. For better clarity, policy $\pi_\psi$ is hereinafter referred to as vocabulary distribution.

---

**Algorithm 1:** Refined ARIL

---

Initialize the vocabulary distribution $\pi_\psi$ and discriminator $f_{\theta,\varphi}$.
**for** iteration $i$ in $\{1, ..., N\}$ **do**
    Obtain caption $\{w_1^{\text{true}}, ..., w_n^{\text{true}}\}$ from the ground truth.
    Collect generated caption $\{w_1, ..., w_n\}$ using the vocabulary distribution $\pi_\psi$.
    $D_{\theta,\varphi} \leftarrow \text{sigmoid}(f_{\theta,\varphi} - \log(\pi_\psi))$
    Update $(\theta, \varphi)$ via Eq. (5) for the discriminator.
    Update $\psi$ via Eq. (16) for the generator.
**end**

---

### 4.1 DISCRIMINATOR

The objective of the discriminator is to distinguish the true caption from the generated one. At time $t$, the discriminator maximizes the divergence in Eq. (4) by

$$L_t(\theta, \varphi) = -\log\left(D_{\theta,\varphi}(w_t^{\text{true}}, s_t^{\text{true}})\right)_{w_t^{\text{true}} \sim p_{\text{true}}} - \log\left(1 - D_{\theta,\varphi}(w_t, s_t)\right)_{w_t \sim \pi_\psi} \tag{5}$$

where $p_{\text{true}}$ is the distribution of the true caption, and $\pi_\psi$ is the vocabulary distribution estimated by the generator. $D_{\theta,\varphi}$ is computed as in Eq. (1) and Eq. (2), where the discriminator learns the reward function $f_{\theta,\varphi}$ for $D_{\theta,\varphi}$ and the generator estimates the vocabulary distribution $\pi_\psi$ for $D_{\theta,\varphi}$, respectively.

### 4.2 GENERATOR

In the following, $D_{\theta,\varphi}$ is represented as below (Fu et al., 2018) using Eq. (1) and Eq. (2):

$$D_{\theta,\varphi}(w_t, s_t) = \text{sigmoid}\left(f_{\theta,\varphi}(w_t, s_t) - \log(\pi_\psi)\right) \tag{6}$$

Given word $w_t$ that is sampled from the vocabulary distribution $\pi_\psi$, the generator maximizes $D_{\theta,\varphi}(w_t, s_t)$ by

$$\begin{aligned}
L_t(\psi) &= -\mathbb{E}_{w_t \sim \pi_\psi}[\log\left(D_{\theta,\varphi}(w_t, s_t)\right) - \log\left(1 - D_{\theta,\varphi}(w_t, s_t)\right)] \\
&= -\mathbb{E}_{w_t \sim \pi_\psi}[f_{\theta,\varphi}(w_t, s_t) - \log(\pi_\psi)]
\end{aligned} \tag{7}$$

Using REINFORCE algorithm (Sutton & Barto, 1998), the gradient $\nabla_\psi L_t$ becomes:

$$\begin{aligned}
\nabla_\psi L_t &= -\sum\nolimits_{\pi_\psi} \left(f_{\theta,\varphi}(w_t, s_t) - \log(\pi_\psi)\right)\nabla_\psi \pi_\psi + \pi_\psi \nabla_\psi \left(f_{\theta,\varphi}(w_t, s_t) - \log(\pi_\psi)\right) \\
&= -\sum\nolimits_{\pi_\psi} \pi_\psi \left[\frac{1}{\pi_\psi}\left(f_{\theta,\varphi}(w_t, s_t) - \log(\pi_\psi)\right)\nabla_\psi \pi_\psi + \nabla_\psi\left(f_{\theta,\varphi}(w_t, s_t) - \log(\pi_\psi)\right)\right] \\
&= -\frac{1}{\pi_\psi}\left(f_{\theta,\varphi}(w_t, s_t) - \log(\pi_\psi)\right)\nabla_\psi \pi_\psi - \nabla_\psi\left(f_{\theta,\varphi}(w_t, s_t) - \log(\pi_\psi)\right) \\
&= -\frac{1}{\pi_\psi}\left(f_{\theta,\varphi}(w_t, s_t) - \log(\pi_\psi) - 1\right)\nabla_\psi \pi_\psi
\end{aligned} \tag{8}$$

When the generator converges (i.e.,$\nabla_\psi L_t = 0$), there exists two stationary points: $\nabla_\psi \pi_\psi = 0$ and $\log(\pi_\psi) = f_{\theta,\varphi}(w_t, s_t) - 1$. If Nash equilibrium can be reached at optimality, $D_{\theta,\varphi}$ should converge to 0.5 when $\nabla_\psi L_t = 0$. Thus it's only possible for the first point to reach Nash equilibrium since $D_{\theta,\varphi} = \text{sigmoid}\left(f_{\theta,\varphi}(w_t, s_t) - \log(\pi_\psi)\right) = \text{sigmoid}(1) \neq 0.5$ at the second point. However, even for the first point, Nash equilibrium exists only for a concave $\pi_\psi$, requiring Hessian of the gradient vector filed being positive definite (Mescheder & Geiger, 2017). To relax this constraint, a *constant term* is added into the expectation in Eq. (7)

$$L_t(\psi) = -\mathbb{E}_{w_t \sim \pi_\psi}[f_{\theta,\varphi}(w_t, s_t) - \log(\pi_\psi) + 1] \tag{9}$$

$$\nabla_\psi L_t = -\frac{1}{\pi_\psi}(f_{\theta,\varphi}(w_t, s_t) - \log(\pi_\psi))\nabla_\psi \pi_\psi \tag{10}$$

to expand the feasible region of reaching Nash equilibrium by shifting the second stationary point to $D_{\theta,\varphi}(w_t) = \text{sigmoid}(0) = 0.5$. The *constant term* makes it possible to achieve Nash equilibrium

even for a non-concave $\pi_\psi$, which also makes it easier for the adversarial model to converge . It is noted that the *constant term* can also be regarded as *baseline* in REINFORCE, except it is utilized to centralize the stationary point instead of reducing variance of the estimation.

In practice, mode collapse occurs when the generator produces a single or limited modes, which exhibits as little diversity in image captioning. To mitigate mode collapse (Mirza & Osindero, 2014) and increase the diversity of the generated captions, we add ground truth data into the generator as a *conditional term*:

$$
\begin{aligned}
L_t(\psi) &= -\mathbb{E}_{w_t \sim \pi_\psi}[f_{\theta,\varphi}(w_t, s_t) - \log(\pi_\psi) + 1] - \mathbb{E}_{w_t^{\text{true}} \sim \pi_\psi^{\text{true}}}[f_{\theta,\varphi}(w_t^{\text{true}}, s_t^{\text{true}}) - \log(\pi_\psi^{\text{true}}) + 1] \\
&= -\big(f_{\theta,\varphi}(w_t, s_t) - \log(\pi_\psi)\big)\log(\pi_\psi) - \big(f_{\theta,\varphi}(w_t^{\text{true}}, s_t^{\text{true}}) - \log(\pi_\psi^{\text{true}})\big)\log(\pi_\psi^{\text{true}})
\end{aligned}
$$
(11)

where $\pi_\psi^{\text{true}}$ is the approximated true caption distribution in the generator, and $\mathbb{E}_{w_t^{\text{true}} \sim \pi_\psi^{\text{true}}}[\cdot]$ is the *conditional term*. See Appendix G for the deduction details. The coefficient of $\log(\pi_\psi^{\text{true}})$ is symmetrical to the coefficient of $\log(\pi_\psi)$ and is updated adaptively in the training process. The *conditional term* helps in strengthening the generator in the adversarial training. When $D_{\text{true}} > D_{\text{gen}}$, the gradient of the true data becomes larger than that of the generated one ($\nabla_{\pi_\psi^{\text{true}}} L_t > \nabla_{\pi_\psi} L_t$), and thus the generator further increases the probability of the true word ($\pi_\psi^{\text{true}}$). Otherwise (i.e., $D_{\text{true}} < D_{\text{gen}}$), the generator prefers sampling its self-generated words to fool the discriminator. By switching between the true words and the generated ones, the generator maintains informative gradient during the adversarial training (Peng et al., 2019). Note that adding the conditional term does not change the model's convergence to Nash equilibrium since $\pi_\psi = \pi_\psi^{\text{true}}$ at the second stationary point.

## 5 EXPERIMENTS

In the experiments, we validate the effectiveness of the proposed algorithm by answering three questions: 1) Is the caption evaluator (i.e., discriminator) capable of learning compact reward? 2) Driven by the learned reward, is the caption generator effective to produce qualitative captions? 3) How does our algorithm perform when built on or compared with existing methods?

To answer 1), we first tested the compactness of the learned reward by observing performance of the collected top-$k$ captions. Then we explored the correlation between the learned reward and the human evaluation results. To answer 2), we evaluated the quality of the generated caption on its content, diversity and grammar. To answer 3), we built our algorithm on existing learning methods and compared their performance. We also conducted ablation experiments to demonstrate the importance of each component of our algorithm.

### 5.1 IMPLEMENTATION DETAILS

We conducted experiments on the well-known benchmark dataset MS COCO (Chen et al., 2015). The dataset has 123,287 labeled images and each image has at least 5 human annotated captions as reference. To assess the robustness of our algorithm, we use two splits of the COCO dataset: standard split (Karpathy & Fei-Fei, 2015) which is created by randomly picking test images, and robust split (Lu et al., 2018) which is organized to maximize difference of the co-occurrence distribution between the training and test set. The robust split is recently proposed and is more challenging due to its distribution difference between the training and test set. The standard split has $113287/5000/5000$ train/val/test images and the robust split has $110234/3915/9138$ train/val/test images.

We implement our algorithm using Adam optimizer (Kingma & Ba, 2014) with fixed learning rate $10^{-5}$. Our vocabulary size is fixed to 10,000 including a special start sign <BOS>and an end sign <EOS>. In the generator, the number of hidden nodes of every layer is $512$. For simplicity, the discriminator has the same model structure as the generator except having one additional layer for estimating $h_\varphi$. For fair comparison, all the methods in MLE, RL, GAN, AIRL and rAIRL were produced using the same image features and model structure in (Anderson et al., 2018). Specifically, RL is the self-critical algorithm in (Rennie et al., 2017). GAN is the adversarial algorithm in (Dai & Lin, 2017) that learns sentence-level reward. AIRL is the standard adversarial inverse reinforcement learning method in (Fu et al., 2018), and rAIRL is the proposed method. Note that our scores of MLE are lower on the standard split but higher on the robust split than (Anderson et al., 2018) because

1) we used fixed number of the bounding box (i.e., 36) for simplicity; 2) the hyperparameters were tuned to adapt to both splits and thus are not exactly the same with (Anderson et al., 2018).

## 5.2 PERFORMANCE OF THE RECOVERED REWARD

Table 1: Correlation between the reward differences and semantic differences by replacing a given word with a similar word (RP_S) and a distinct word (RP_D), respectively.

| Method | Standard Split | | Robust Split | |
|---|---|---|---|---|
| | RP_S | RP_D | RP_S | RP_D |
| RL | 0.03 | 0.01 | -0.10 | 0.00 |
| GAN | 0.07 | 0.21 | 0.04 | 0.18 |
| AIRL | 0.15 | 0.11 | 0.30 | 0.20 |
| rAIRL | **0.54** | **0.30** | **0.51** | **0.31** |

**Compactness.** Compactness means that the reward values should be close for similar words and different for distinct words. For example, *kid* can also be referred to as *little boy* or *little girl*, and thus their reward values should be close to each other in the discriminator. To see the correlation between the reward differences and semantic differences, we replace a given word $w_t$ in the generated caption with a similar word $w_{similar}$ and a distinct word $w_{distinct}$, respectively. Specifically, in a generated caption, the first word that belongs to the COCO 80 class[1] (Lu et al., 2018) is replaced. A sentence is discarded if no word can be replaced. The words within the same class are considered to be similar (such as *bike* and *bicycle*), and the words that belong to difference classes are distinct (such as *man* and *bike*). For RL, since it maximizes a handcrafted reward (SPICE (Anderson et al., 2016) in our experiment) instead of learning a reward function, the reward difference is the variation of SPICE before and after replacement. For reward-learning methods, the reward difference is the variation of the learned reward. The semantic difference is the Euclidean distance between the Glove embedding vectors of two words (Pennington et al., 2014). The results are reported in Table 1. Higher correlation indicates better compactness. RL serves as a baseline in that the handcrafted reward SPICE compares $n$-gram overlapping without considering the semantic difference. The reward differences of rAIRL correlate the best with the semantic differences for both similar words and distinct words, proving the compactness of the learned reward. It's also noted that due to the reward ambiguity problem, the reward differences of GAN poorly correlate with the semantic differences, especially for similar words.

Table 2: Sentence-level correlation with human evaluation. All p-value (not shown) are less than 0.001.

| Method | Correctness | | | Throughness | | |
|---|---|---|---|---|---|---|
| | Pearson | Spearman | Kendall | Peason | Spearman | Kendall |
| SPICE | 0.44 | 0.45 | 0.39 | 0.45 | 0.46 | 0.38 |
| GAN | 0.12 | 0.11 | 0.15 | 0.12 | 0.11 | 0.15 |
| AIRL | 0.04 | 0.06 | 0.08 | 0.05 | 0.06 | 0.07 |
| rAIRL | 0.43 | 0.40 | 0.35 | 0.40 | 0.37 | 0.34 |
| rAIRL+SPICE | **0.47** | **0.46** | **0.41** | **0.46** | **0.47** | **0.39** |

**Correlation with human evaluation.** As a caption evaluator, the discriminator learns $g_\theta$ that recovers the true reward up to a constant at optimality (Fu et al., 2018). We explore the correlation between the recovered reward and the human evaluation scores from the COMPOSITE dataset (Aditya et al., 2017), where the Amazon Mechanical Turk (AMT) workers evaluate two aspects of the captions (i.e., correctness and throughness) at a range of 1-5, see Appendix C for details of the human evaluation process. The correlation is evaluated using Pearson $p$, Kendall's $\tau$ and Spearman's $r$ correlation coefficients. In Table 2, the reward of AIRL/rAIRL is the sum of the word-wise reward $g_\theta$, and the reward of rAIRL+SPICE is a linear combination of $g_\theta$ and the SPICE score. Among the reward-learning methods, AIRL poorly correlates with human, whereas the proposed rAIRL improves AIRL on all the correlation metrics, especially on the Pearson correlation (from $0.04$ to $0.43$). Furthermore, a simple combination of SPICE and the recovered reward leads to an increased correlation with the human scores, which proves the capacity of the discriminator as a caption evaluator.

---

[1]https://github.com/jiasenlu/NeuralBabyTalk/blob/master/data/coco/coco_class_name.txt

## 5.3 EVALUATION ON THE GENERATED CAPTIONS.

Table 3: Evaluation scores on generated captions. The best score is in bold font and the second best score is underlined. SPICE is the handcrafted evaluation metric. CHAIR$_s$ and CHAIR$_i$ represent the object hallucination ratio at sentence level and instance level, respectively. HE indicates human evaluation. VC indicates vocabulary coverage and NS is the ratio of novel sentences.

| Method | Standard Split | | | | | | Robust Split | | | | | |
|---|---|---|---|---|---|---|---|---|---|---|---|---|
| | SPICE | CHAIR$_s$ | CHAIR$_i$ | HE | VC | NS | SPICE | CHAIR$_s$ | CHAIR$_i$ | HE | VC | NS |
| MLE | 19.0 | 8.3 | 6.0 | 16.1 | 12.4 | 49.7 | 18.6 | 19.1 | 16.9 | 18.0 | 12.5 | 58.8 |
| RL | **20.7** | 11.4 | 8.5 | 8.7 | 11.4 | **88.5** | 18.1 | 25.2 | 20.4 | 6.4 | 12.7 | **87.3** |
| GAN | 18.3 | 7.6 | 6.4 | 19.9 | 13.4 | 75.0 | 16.8 | 17.3 | 15.2 | 20.2 | 15.3 | 75.6 |
| AIRL | 17.3 | 12.7 | 10.3 | 14.0 | 12.3 | 67.3 | 16.7 | 22.7 | 18.5 | 14.8 | 15.6 | 73.8 |
| rAIRL | 20.4 | **7.2** | **5.5** | **41.3** | **13.6** | 76.1 | **18.7** | **17.1** | **14.3** | **40.6** | 15.8 | 76.5 |

**Content correctness.** For a comprehensive evaluation of the content correctness, the results of both the handcrafted metrics and human studies are shown in Table 3. For the handcrafted metrics, we report scores of SPICE and the recently proposed CHAIR$_s$ and CHAIR$_i$ since they correlate well with human (Anderson et al., 2016; Rohrbach et al., 2018). SPICE computes similarity with the ground truth captions based on scene graph whilst CHIAR$_s$ and CHIAR$_i$ indicate ratio of hallucinated objects. The full results of other handcrafted metrics can be found in Appendix E. Compared with non-adversarial methods (i.e., MLE, RL), existing adversarial net (GAN) does not perform well on SPICE due to the reward ambiguity problem, whereas our rAIRL improves GAN (from 16.8 to 18.7) by disentangling reward for each word, and even outperforms RL (from 18.1 to 18.7) on the robust split. The lowest scores on CHIAR$_s$ and CHIAR$_i$ suggest that object hallucination is less likely in rAIRL. As for the human evaluation, HE in Table 3 indicates the percentage of captions that are considered the best among the five methods. See Appendix C for details of the human evaluation process. The descriptions generated by our rAIRL are considered the best for over $40\%$ images, whilst RL has the lowest scores that are less than $10\%$.

**Diversity.** The diversity of captions is evaluated on a corpus level, indicating to what extent the generated captions of different images have diverse expressions. The results are presented in Table 3. VC indicates vocabulary coverage, which is the number of vocabularies of the generated captions over number of vocabularies of the ground truth captions. NS represents ratio of novel sentence, which is the ratio of sentences that do not appear in the training set. The fact that RL has high ratio of novel sentence ($88.5\%/87.3\%$) but low vocabulary coverage ($11.4\%/12.7\%$) suggests that it uses high-frequency words (such as "in a", "of a") to reconstruct captions that appear to be different from the training set (Li et al., 2019a). See Appendix A for a few examples. Our rAIRL improves AIRL on the diversity metrics and outperforms other learning methods on vocabulary coverage, indicating its capability of generating diverse descriptions on a corpus level.

Table 4: Percentage of different grammar errors found in the generated captions. Re represents Redundancy, AE is Agreement Error, AM denotes Article Misuse and IS is Incomplete Sentence.

| Method | Standard Split | | | | | Robust Split | | | | |
|---|---|---|---|---|---|---|---|---|---|---|
| | Total | Re | AE | AM | IS | Total | Re | AE | AM | IS |
| MLE | 0.78 | 0.04 | 0.56 | 0.14 | 0.04 | **0.57** | 0.04 | 0.26 | 0.16 | 0.10 |
| RL | 5.64 | 0.90 | 0 | 3.36 | 1.38 | 4.67 | 0.19 | 0.02 | 3.8 | 0.69 |
| GAN | 1.24 | 0.62 | 0.18 | 0.06 | 0.38 | 2.40 | 1.10 | 0.40 | 0.26 | 0.63 |
| AIRL | 1.68 | 0.04 | 0.62 | 0.70 | 0.32 | 1.20 | 0.10 | 0.27 | 0.72 | 0.12 |
| rAIRL | **0.75** | 0.14 | 0.20 | 0.21 | 0.20 | **0.57** | 0.14 | 0.17 | 0.16 | 0.10 |

**Grammar.** We used LanguageTool [2] to check grammar of the generated captions. Table 4 shows percentage of sentences that have grammar errors found by LanguageTool: 1) *Redundancy* means repeated phrases in a sentence; 2) *Agreement Error* means subject-verb agreement error, such as "people is"; 3) *Article Misuse* denotes inappropriate usage of indefinite articles, such as using "a" before uncountable nouns or plural words; 4) *Incomplete Sentence* refers to incomplete sentence that lacks a subject. We found captions produced by RL have the most grammar errors ($5.64\%$ on the standard split and $4.67\%$ on the robust split), especially the Article Misuse. On the other hand, by approximating the true data distribution of each word in the sentence, rAIRL and MLE have the least grammar errors among all learning methods ($0.75\%/0.78\%$ on the standard split and $0.57\%/0.57\%$

---

[2] https://languagetool.org

on the robust split)). We also noticed that each method except rAIRL is biased towards a particular type of grammar error: agreement error in MLE, article misuse in RL, redundancy in GAN, article misuse in AIRL. On both splits, our rAIRL does not appear to be biased towards a specific type of these grammar errors.

**Summary.**    The proposed rAIRL constantly performs well on both splits of MS COCO and is capable of producing qualitative captions with few grammar errors. As a new adversarial algorithm, rAIRL enhances GAN by disentangling compact reward for each word in the caption and improves AIRL by shifting the stationary point towards Nash equilibrium. In the following sections, we first give ablation studies to see which component of our method explains the performance improvements, and then compare rAIRL with existing methods.

## 5.4 Comparison Results

Table 5: Ablation methods of rAIRL. "term1" is the constant term in Eq. (9) and "term2" is the conditional term in Eq. (16). GE denotes grammar error rate.

| Method | Standard Split | | | | | | Robust Split | | | | | |
|---|---|---|---|---|---|---|---|---|---|---|---|---|
| | SPICE | CHAIR$_s$ | CHAIR$_i$ | VC | NS | GE | SPICE | CHAIR$_s$ | CHAIR$_i$ | VC | NS | GE |
| rAIRL(w/o term1) | 18.8 | 10.5 | 8.2 | 12.8 | 73.5 | 1.07 | 17.0 | 19.9 | 17.5 | 14.1 | 71.6 | 0.95 |
| rAIRL(w/o term2) | 19.3 | 9.4 | 7.4 | 12.2 | 71.3 | 0.83 | 17.9 | 18.9 | 15.8 | 13.7 | 62.4 | 0.72 |
| rAIRL | **20.4** | **7.2** | **5.5** | **13.6** | **76.1** | **0.75** | **18.7** | **17.1** | **14.3** | **15.8** | **76.5** | **0.57** |

**Ablation studies.**    We conducted ablation experiments to understand the importance of each component of our algorithm. Specifically, the *constant term* in Eq. (9) and the *conditional term* in Eq. (16) is removed, respectively. Scores of all the evaluation techniques mentioned above are presented in Table 5. We found that all the scores drop after removing either one of the terms. Comparing these two terms, the *constant term* seems more important in recognizing objects and relations in the image since removing it has larger drop on SPICE. The lager drop on vocabulary coverage and ratio of novel sentence in the second row indicates that the *conditional term* plays a significant role in increasing the diversity of the generated captions. More results on using different model architectures are included in Appendix D.

Table 6: Comparison with existing methods on the handcrafted evaluation metrics.

| Learning Method | Model | Standard Split | | | Robust Split | | |
|---|---|---|---|---|---|---|---|
| | | BLEU4 | CIDEr | SPICE | BLEU4 | CIDEr | SPICE |
| MLE | Att2in | 31.3 | 101.3 | - | 31.5 | 90.6 | 17.7 |
| | NBT | 34.7 | 107.2 | 20.1 | 31.7 | 94.1 | 18.3 |
| | Up-Down | **36.2** | **113.5** | 20.3 | **31.6** | 92.0 | 18.1 |
| | rAIRL+MLE(Up-Down) | 34.6 | 112.9 | **20.7** | 31.1 | **96.8** | **19.1** |
| RL | GAN$_2$(SCST, Co-att, log(D)+5×CIDEr) | - | 111.1 | - | - | - | - |
| | Att2in | 33.3 | 111.4 | - | - | - | - |
| | Up-Down | **36.3** | **120.1** | **21.4** | - | - | - |
| | rAIRL+RL(Up-Down) | 35.0 | 115.7 | 21.3 | 30.8 | 97.9 | 19.7 |
| GAN | G-GAN | 20.7 | 79.5 | 18.2 | - | - | - |
| | GAN$_3$ (SCST, Co-att, log(D)) | - | 97.5 | - | - | - | - |
| | rAIRL(Up-Down) | **33.8** | **110.2** | **20.4** | 30.2 | 93.7 | 18.7 |

**Comparison with existing methods.**    Based on the learning methods, existing models are divided into three categories in Table 6, and we chose recent proposed methods for comparison: Att2in (Rennie et al., 2017), G-GAN (Dai & Lin, 2017), NBT (Lu et al., 2018), Up-Down (Anderson et al., 2018) and GAN$_2$, GAN$_3$ (Dognin et al., 2019). Although some metrics based on $n$-gram overlapping (BLEU4, CIDEr) do not correlate well with human, their results are also reported in Table 6 for fair comparison. Among the adversarial methods (GAN category), our rAIRL performs the best on all metrics. The results on COCO online server are given in Appendix E.

To further demonstrate the generalization ability of our algorithm, we built our algorithm on the non-adversarial based models. The composite models are denoted with rAIRL+MLE and rAIRL+RL. In rAIRL+MLE, the conditional term is replaced by the cross-entropy loss of MLE; in rAIRL+RL, the RL loss is added into the loss function of the generator. In Table 6, our rAIRL+MLE further improves the MLE baseline (i.e., Up-Down using MLE loss) on SPICE, whereas rAIRL+RL does not improve the RL baseline (i.e., Up-Down using RL loss) on these evaluation metrics. This is caused by the difficulty of normalizing the learned reward and the handcrafted reward to the same order of magnitude (Shelton Christian, 2001), and we leave this problem to our future work.

## 6 CONCLUSION

In this paper, we address the reward ambiguity problem in image captioning and propose a refined Adversarial Inverse Reinforcement Learning (rAIRL) method that solves the problem by disentangling reward for each word in a sentence. Moreover, it achieves stable adversarial training by refining the loss function to shift the stationary point towards Nash equilibrium, and mode control technique is incorporated to mitigate mode collapse. It is demonstrated that our method can learn compact reward through extensive experiments on MS COCO.

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

# A Visualized Results of Generated Captions

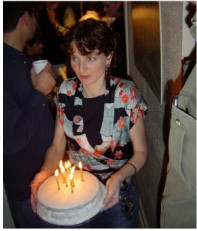

**MLE:** a girl is looking at a birthday cake. (8.7)
**RL:** a woman is holding a cake in a room. (17.4)
**GAN:** a woman getting out candles on a birthday cake. (33.0)
**rAIRL:** a woman **holding** a cake with **candles** on it (36.4)

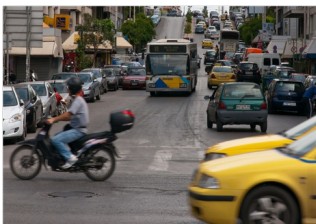

**MLE:** a city street filled with lots of traffic. (18.2)
**RL:** a yellow bus driving down a city street. (18.2)
**GAN:** a man riding a motorcycle on a street. (22.9)
**rAIRL:** a **man** riding a motorcycle down a **busy street**. (25.2)

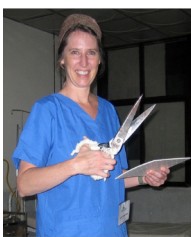

**MLE:** a woman is holding a pair of scissors. (23.1)
**RL:** a woman holding a pair of scissors in a. (26.7)
**GAN:** a woman holding a pair of scissors. (23.1)
**rAIRL:** a **smiling** woman holding a pair of scissors. (26.7)

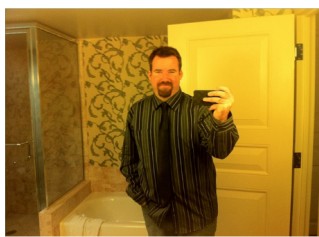

**MLE:** a man taking a picture of himself in a mirror. (30.5)
**RL:** a man standing in a bathroom with a toilet. (23.5)
**GAN:** a man standing in front of a mirror. (26.7)
**rAIRL:** a man taking a **selfie** in a **bathroom mirror**. (44.4)

Figure 2: Captions produced by different methods from the test set (standard split). Beside each caption we report SPICE score. Captions generated by rAIRL are correct and human-like in these examples.

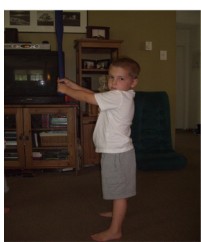

**MLE:** a young boy standing in front of a tv. (27.5)
**RL:** a young boy playing a video game in a living room. (7.7)
**GAN:** a little boy standing in a living room. (34.5)
**rAILR:** a young boy is playing a video game. (14.3)

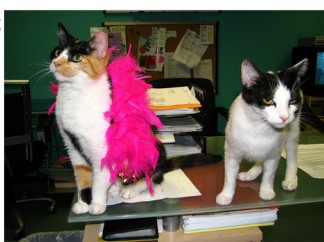

**MLE:** a black and white cat sitting on a desk. (30.6)
**RL:** a white cat sitting on top of a desk with a. (21.4)
**GAN:** a cat that is sitting on a desk. (16.0)
**rAIRL:** a cat sitting on top of a computer desk. (22.2)

Figure 3: Failed examples of rAIRL. The objects and relations are not correctly recognized in these pictures.

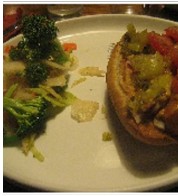

**MLE:** a closeup of a plate of food.
**RL:** a plate of food on a table with a plate.
**GAN**: a plate of food on a plate.
**AIRL:** a plate of food is on a plate.
**rAIRL:** a plate of food on a table.

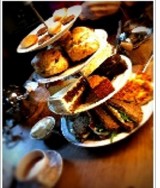

**MLE:** a closeup of plates of food on a table.
**RL:** a plate of food on a table with a plate.
**GAN:** a lunch of food on a plate.
**AIRL:** a plate of food on a plate.
**rAIRL:** a table **topped with plates of food** on it.

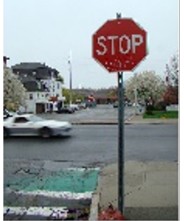

**MLE:** a stop sign on the side of a road.
**RL:** a stop sign on the side of a street.
**GAN:** a stop sign on a street corner
**AIRL:** a stop sign on the side of a street.
**rAIRL:** a red stop sign sitting on the side of a road.

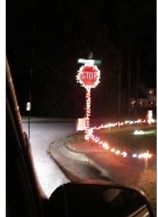

**MLE:** a clock on a street at night at night.
**RL:** a traffic light on the side of a street.
**GAN:** a stop sign on the street.
**AIRL:** a view of a stop sign on a street.
**rAIRL:** a view of a stop sign on **a lit street at night.**

Figure 4: Examples showing diversity of the captions. The left and right columns show pictures with similar content but different details. The proposed rAIRL successfully recognizes these differences and gives diverse captions.

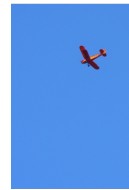 1) an airplane is flying in the blue sky.
2) a plane is flying in the blue sky.
3) an airplane flying through the blue sky.
4) an airplane is flying in the air.
5) a red plane is flying in the sky.

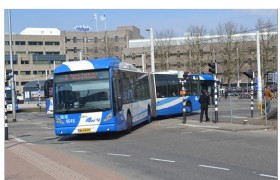 1) a blue and white bus driving down a street.
2) a blue and white bus on a city street.
3) a public transit bus on a city street.
4) a city bus is driving down the street.
5) a bus driving down a city street.

Figure 5: Examples of the top-5 generated captions of rAIRL. Driven by the compact reward function, the generator describes a given image with semantically similar words.

## B  DIAGNOSE AND IMPROVE CAPTIONS

Table 7: Results of rewriting caption from the located position by rAIRL on MS COCO standard split. Beside each score we report its improvement relative to rewriting from a random position.

| Source Caption | BLEU1 | | BLEU2 | | BLEU3 | | BLEU4 | | ROUGE_L | | CIDEr | | SPICE | |
|---|---|---|---|---|---|---|---|---|---|---|---|---|---|---|
| | score | Δ | score | Δ | score | Δ | score | Δ | score | Δ | score | Δ | score | Δ |
| MLE | 73.3 | (+0.0) | 57.1 | (-0.2) | 43.4 | (+0.1) | 32.6 | (+0.5) | 51.4 | (+0.1) | 108.5 | (+2.0) | 20.6 | (+0.1) |
| RL | 72.9 | (-0.1) | 56.7 | (+0.3) | 42.4 | (+0.5) | 31.1 | (+0.4) | 50.8 | (+0.2) | 104.0 | (-0.5) | 20.1 | (+0.0) |
| GAN | 72.7 | (+1.1) | 56.3 | (+1.2) | 42.0 | (+1.0) | 30.9 | (+0.8) | 50.6 | (+0.4) | 103.0 | (+2.3) | 20.0 | (+0.4) |
| AIRL | 72.6 | (+1.0) | 56.1 | (+1.3) | 41.8 | (+1.2) | 30.5 | (+0.9) | 50.6 | (+0.8) | 102.6 | (+4.1) | 19.9 | (+0.8) |

Since the proposed rAIRL learns a word-wise reward, it's also applicable to diagnose a given caption by finding the wrong word (e.g., the word whose reward decreases sharply compared with that of the previous word) and rewriting the caption to improve its quality. For example, improving *a man is playing soccer* to *a man and a kid are playing soccer* can be done by rewriting the caption from *is*. Therefore, we choose to rewrite a given caption (source caption) from the word whose reward has a decrease rate larger than $50\%$. However, we found that even rewriting the source caption from a random position using rAIRL can also improve the evaluation scores. Thus, rewriting from a random position is selected as the baseline to compare with rewriting from the located position. Table 7 shows results of rewriting from the located position, where the source captions are given by MLE, RL, GAN and AIRL. Beside each score we report its improvement relative to rewriting from a random position, whose values are mostly positive. This demonstrates that the proposed rAIRL can diagnose the caption at a word level, and further improves the caption quality by rewriting from the located position.

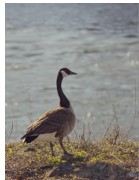 **Before:** a black and white bird standing in the sand.

**After:** a black and white bird standing on the edge of a lake.

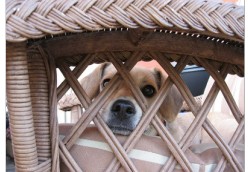 **Before:** a dog sitting on top of a chair.

**After:** a dog is looking out of a wicker chair.

Figure 6: Examples showing the generated captions from AIRL before and after re-written.

## C  HUMAN EVALUATION PROCESS

We conducted two types of human studies, one for evaluating the learned reward (in Section 5.2), and the other for examining quality of the generated captions (in Section 5.3). In the first human study experiment (in Section 5.2), we used the human scores in the COMPOSITE[3] dataset (Aditya et al., 2017), whose images are subsets from Flickr8k, Flickr30k and MS COCO. The descriptions from this dataset are either ground truth captions or generated sentences by (Aditya et al., 2017; Johnson et al., 2015). In the human evaluation process, the AMT worker was asked to give a score at range of

---

[3]https://imagesdg.wordpress.com/image-to-scene-description-graph/

Table 8: Full results of the sentence-level correlation. All p-value (not shown) are less than 0.001.

| Method | Correctness | | | Throughness | | |
|---|---|---|---|---|---|---|
| | Peason | Spearman | Kendall | Peason | Spearman | Kendall |
| BLEU1 | 0.19 | 0.27 | 0.19 | 0.20 | 0.28 | 0.20 |
| BLEU4 | 0.33 | 0.30 | 0.22 | 0.32 | 0.31 | 0.22 |
| CIDEr | 0.40 | 0.45 | 0.37 | 0.41 | 0.45 | 0.36 |
| SPICE | 0.44 | 0.45 | 0.39 | 0.45 | 0.46 | 0.38 |
| GAN | 0.12 | 0.11 | 0.15 | 0.12 | 0.11 | 0.15 |
| AIRL | 0.04 | 0.06 | 0.08 | 0.05 | 0.06 | 0.07 |
| rAIRL | 0.43 | 0.40 | 0.35 | 0.40 | 0.37 | 0.34 |
| rAIRL+BLEU1 | 0.44 | 0.41 | 0.35 | 0.41 | 0.39 | 0.34 |
| rAIRL+BLEU4 | 0.45 | 0.43 | 0.36 | 0.42 | 0.42 | 0.35 |
| rAIRL+CIDEr | 0.43 | 0.45 | 0.38 | 0.42 | 0.46 | 0.37 |
| rAIRL+SPICE | **0.47** | **0.46** | **0.41** | **0.46** | **0.47** | **0.39** |

1-5 to evaluate the correctness and throughness of each sentence. Captions with length exceeding 20 were discarded, resulting a total of $11,657$ sentences. Full results of the correlation is shown in Table 8. SPICE correlates better with human evaluation when compared with other handcrafted metrics, whilst the composite metric rAIRL+SPICE further increases the correlation.

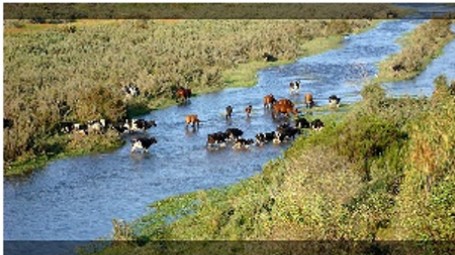

1: A herd of cattle drinking from a pond.
2: A herd of cows in the water.
3: A group of cows standing by a river.
4: A herd of cows grazing in the water.
5: A herd of cattle drinking from a river.

Figure 7: An example of the images shown to the human evaluator. The captions were produced by MLE, GAN, RL, AIRL and rAIRL methods in a randomized order.

In the second human study experiment (in Section 5.3), we randomly selected 500 test images from the standard split and robust split of MS COCO, respectively. The worker was asked "which caption is the best" by given an image with five sentences generated from the adversarial and non-adversarial methods, as shown in Figure 7. The worker was allowed but not encouraged to make multiple choices if he/she thinks these captions are equally correct. The order of captions produced by different methods was randomized. Following (Shetty et al., 2017), each image in the test set was evaluated by 5 workers.

## D   ABLATION EXPERIMENTS ON MODEL ARCHITECTURES

Theoretically, our algorithm is model-agnostic since it is independent of the design of model architecture. For empirical support of the claim, we show results of using Att2in (Rennie et al., 2017) and Up-Down (Anderson et al., 2018) architectures in Table 9. We report the metrics used in the original paper for fair comparison. The proposed rAIRL mainly improves SPICE, which correlates well with human evaluations, on both architectures.

Table 9: Results of using different model architectures in our method.

| Method | Standard Split | | | Robust Split | | |
|---|---|---|---|---|---|---|
| | BLEU4 | CIDEr | SPICE | BLEU4 | CIDEr | SPICE |
| Att2in | 31.0 | 101.3 | - | **31.5** | 90.6 | 17.7 |
| rAIRL(Att2in) | **31.3** | **105.2** | **19.9** | 30.7 | **92.5** | **18.0** |
| Up-Down | **36.2** | **113.5** | 20.3 | **31.6** | 92.0 | 18.1 |
| rAIRL(Up-Down) | 33.8 | 110.2 | **20.4** | 30.2 | **93.7** | **18.7** |

# E  FULL RESULTS ON MS COCO

We adopt SPICE to evaluate content correctness in the paper because it has better correlation with human judgments(Anderson et al., 2016). Table 10 gives full results of the handcrafted metrics on two splits of MS COCO. Comparing the adversarial (GAN, AIRL, rARIL) and non-adversarial (MLE, RL) methods, RL outperforms other methods on most metrics. In adversarial methods, the proposed rAIRL performs the best. Table 11 shows results on the MS COCO online test server. The proposed rAIRL improves AIRL on all the metrics.

Table 10: Results of the conventional handcrafted metrics on MS COCO test split.

| Method | Standard Split | | | | | | | Robust Split | | | | | | |
|---|---|---|---|---|---|---|---|---|---|---|---|---|---|---|
| | BLEU1 | BLEU2 | BLEU3 | BLEU4 | ROUGE_L | CIDEr | SPICE | BLEU1 | BLEU2 | BLEU3 | BLEU4 | ROUGE_L | CIDEr | SPICE |
| MLE | 74.5 | 57.0 | 41.7 | 30.3 | 50.2 | 104.6 | 19.0 | 69.5 | 52.7 | 39.2 | 29.3 | 48.7 | 93.4 | 18.6 |
| RL | 75.5 | 58.2 | 43.8 | 35.0 | 51.6 | 115.1 | 20.7 | 73.5 | 55.9 | 40.8 | 29.9 | 49.9 | 95.1 | 18.1 |
| GAN | 67.7 | 51.9 | 38.6 | 28.3 | 48.7 | 93.4 | 18.3 | 64.7 | 48.0 | 34.5 | 24.6 | 46.2 | 78.3 | 16.8 |
| AIRL | 69.9 | 53.6 | 39.1 | 27.5 | 49.8 | 87.4 | 17.3 | 67.6 | 50.5 | 36.3 | 25.9 | 47.3 | 79.5 | 16.7 |
| rAIRL | 73.8 | 58.2 | 44.6 | 33.8 | 52.1 | 110.2 | 20.4 | 70.3 | 54.1 | 40.5 | 30.2 | 49.4 | 93.7 | 18.7 |

Table 11: Results on COCO test server. Methods marked with $*$ adopt RL of CIDEr optimization.

| Method | BLEU1 | | BLEU2 | | BLEU3 | | BLEU4 | | METEOR | | ROUGE_L | | CIDEr | |
|---|---|---|---|---|---|---|---|---|---|---|---|---|---|---|
| | c5 | c40 | c5 | c40 | c5 | c40 | c5 | c40 | c5 | c40 | c5 | c40 | c5 | c40 |
| Adaptive (Lu et al., 2017) | 74.8 | 92.0 | 58.4 | 84.5 | 44.4 | 74.4 | 33.6 | 63.7 | 26.4 | 35.9 | 55.0 | 70.5 | 104.2 | 105.9 |
| Att2all$*$ (Rennie et al., 2017) | 78.1 | 93.7 | 61.9 | 86.0 | 47.0 | 75.9 | 35.2 | 64.5 | 27.0 | 35.5 | 56.3 | 70.7 | 114.7 | 116.7 |
| Up-Down$*$(Anderson et al., 2018) | 80.2 | 95.2 | 64.1 | 88.8 | 49.1 | 79.4 | 36.9 | 68.5 | 27.6 | 36.7 | 57.1 | 72.4 | 117.9 | 120.5 |
| AIRL(Up-Down) | 72.5 | 90.4 | 55.2 | 83.7 | 42.3 | 73.6 | 30.8 | 62.6 | 25.0 | 34.2 | 53.5 | 68.5 | 81.9 | 82.6 |
| rAIRL(Up-Down) | 75.4 | 93.1 | 59.8 | 86.5 | 45.9 | 77.2 | 35.0 | 66.7 | 26.1 | 35.4 | 55.6 | 71.1 | 104.1 | 105.2 |
| rAIRL+MLE(Up-Down) | 75.5 | 93.3 | 59.8 | 86.7 | 46.2 | 77.4 | 35.4 | 67.0 | 26.5 | 36.0 | 55.8 | 71.5 | 105.9 | 106.2 |
| rAIRL+RL(Up-Down) $*$ | 79.5 | 94.1 | 63.5 | 88.0 | 48.3 | 78.9 | 36.2 | 68.5 | 27.5 | 36.6 | 56.2 | 71.8 | 112.3 | 115.1 |

# F  DISCUSSION ON LOSS FUNCTIONS

Table 12: Formulas of different loss functions.

| Method | Loss Function |
|---|---|
| MLE | $-\sum_{t=1}^{n} \log(\pi_t^{\text{true}})$ |
| RL | $-r \sum_{t=1}^{n} \log(\pi_t)$ |
| GAN (generator) | $-D_{\text{gen}} \sum_{t=1}^{n} \log(\pi_t)$ |
| rAIRL (generator) | $-\sum_{t=1}^{n} \sigma^{-1}(D_t^{\text{gen}}) \log(\pi_t) - \sigma^{-1}(D_t^{\text{true}}) \log(\pi_t^{\text{true}})$ |

We compare the formula of the proposed loss function with existing methods in Table 12, including MLE, RL and GAN. $n$ is the length of a sentence. $r$ is the handcrafted metric, such as BLEU, CIDEr and SPICE. $\pi_t$ is the probability of the $t_{\text{th}}$ generated word, and $\pi_t^{\text{true}}$ is the probability of the $t_{\text{th}}$ true

word. The loss functions are rewritten using similar symbols for direct comparison. MLE maximizes the probability of the true data $\pi_t^{\text{true}}$ whist RL and GAN optimize the reward by sampling from $\pi_t$. GAN is different from RL in that its reward is learned from the discriminator adversarially instead of being predefined. GAN is capable of mimicking human-written captions by adversarial learning, but the estimated reward function $D_{\text{gen}}$ of a full trajectory can be explained by multiple optimal policies and thus is too ambiguous. The proposed rAIRL further disentangles the reward into $D_t^{\text{gen}}$ at each time step $t$, as well as incorporating the true data for better diversity. From the perspective of loss functions, rAIRL can be regarded as an integration of GAN and MLE using coefficients $\sigma^{-1}(D_t^{\text{gen}})$ and $\sigma^{-1}(D_t^{\text{true}})$.

## G  GENERATOR OBJECTIVE

Below we show the deductions details of the generator objective (i.e., Eq. (16) in section 4.2).

Taking the integral of $\nabla_\psi \mathbb{E}_{w_t \sim \pi_\psi}[\cdot]$ w.r.t. $\psi$

$$\mathbb{E}_{w_t \sim \pi_\psi}[f_{\theta,\varphi}(w_t, s_t) - \log(\pi_\psi) + 1] = \int_\psi \nabla_\psi \mathbb{E}_{w_t \sim \pi_\psi}[f_{\theta,\varphi}(w_t, s_t) - \log(\pi_\psi) + 1]d\psi \quad (12)$$

According to Eq. (10),

$$\nabla_\psi \mathbb{E}_{w_t \sim \pi_\psi}[f_{\theta,\varphi}(w_t, s_t) - \log(\pi_\psi) + 1] = \frac{1}{\pi_\psi}(f_{\theta,\varphi}(w_t, s_t) - \log(\pi_\psi))\nabla_\psi \pi_\psi \quad (13)$$

Thus we have

$$\mathbb{E}_{w_t \sim \pi_\psi}[f_{\theta,\varphi}(w_t, s_t) - \log(\pi_\psi) + 1] = \int_\psi \frac{1}{\pi_\psi}(f_{\theta,\varphi}(w_t, s_t) - \log(\pi_\psi))\nabla_\psi \pi_\psi d\psi$$
$$= \left(f_{\theta,\varphi}(w_t, s_t) - \log(\pi_\psi)\right)\log(\pi_\psi) \quad (14)$$

Similarly, replacing $w_t$ with $w_t^{\text{true}}$ we get

$$\mathbb{E}_{w_t^{\text{true}} \sim \pi_\psi^{\text{true}}}[f_{\theta,\varphi}(w_t^{\text{true}}, s_t^{\text{true}}) - \log(\pi_\psi^{\text{true}}) + 1] = \left(f_{\theta,\varphi}(w_t^{\text{true}}, s_t^{\text{true}}) - \log(\pi_\psi^{\text{true}})\right)\log(\pi_\psi^{\text{true}}) \quad (15)$$

Thus, the generator objective is:

$$L_t(\psi) = -\mathbb{E}_{w_t \sim \pi_\psi}[f_{\theta,\varphi}(w_t, s_t) - \log(\pi_\psi) + 1] - \mathbb{E}_{w_t^{\text{true}} \sim \pi_\psi^{\text{true}}}[f_{\theta,\varphi}(w_t^{\text{true}}, s_t^{\text{true}}) - \log(\pi_\psi^{\text{true}}) + 1]$$
$$= -\left(f_{\theta,\varphi}(w_t, s_t) - \log(\pi_\psi)\right)\log(\pi_\psi) - \left(f_{\theta,\varphi}(w_t^{\text{true}}, s_t^{\text{true}}) - \log(\pi_\psi^{\text{true}})\right)\log(\pi_\psi^{\text{true}})$$
$$(16)$$

