# OpenReview forum: "Learning Compact Reward for Image Captioning"
_ICLR.cc/2020/Conference — Reject_

### Official Review · AnonReviewer1 · 2019-10-16
**Official Blind Review #1**

**Rating:** 6

**Review:**

Overall this is a good paper.
Such compact rewards over image captions are useful for evaluation and diagnosis.

*concerns:
1. method:
   1.1 At first, I think section 4 could be re-written to corporate more concepts in image captioning (such as replacing action a to word w) so that it becomes more readable for related readers.
   1.2 The motivation for adding a constant to shift nash equailibrium could be stated clearer.
   1.3 what is the state s in the context of image captioning? the hidden vector h ?
   1.4 since the reward is computed for each pair of (a, s), how to get the reward for the whole sentence? Sum them up? I wonder this because it seems you compute this in Table 1.
   1.5 in Eq.(5) and Eq.(11), there are expectations. Do you need to compute them using Monte Carlo Rollouts? Or just averaging over mini-batchs? Because G-GAN (Dai et al) also utilizes rollouts to estimate different values for different words.

2. experiments:
  1.1 the experiment in terms of "compactness" may not  reflect well the concept of compactness. While compactness means similar words may obtain similar rewards in the same caption,  top-k captions for an image may be different in multiple words, or different in formats rather than word choices. How about top-k words in a given position of a given caption? Or replace a given word with different words and compare the correlation between reward differences and semantic differences.
  1.2  qualitative samples in terms of caption diversity could be included.
  1.3 since the rewards are computed for each word separately,  the authors may include experiments on using such rewards to diagnose captions, such as replacing a word to make the caption better, etc.

**Experience Assessment:**

I have published in this field for several years.

**Review Assessment: Checking Correctness Of Derivations And Theory:**

I assessed the sensibility of the derivations and theory.

**Review Assessment: Checking Correctness Of Experiments:**

I carefully checked the experiments.

**Review Assessment: Thoroughness In Paper Reading:**

I read the paper thoroughly.

---

> ### Author Response · Authors · 2019-11-12
> **Response to Review #1**
>
> Thank you for your important suggestions and we appreciate your insightful feedback. We have revised our paper accordingly and uploaded the draft. Below please find our response to your suggestions and questions.
>
> 1
> 1.1 As suggested, more related concepts in image captioning are incorporated in section 4 in the revised paper (such as indicating that ${s_t}$ is the hidden state vector, and referring to ${\pi}_{\psi}$ as the vocabulary distribution). Action ${a_t}$ is replaced by word ${w_t}$ for better clarity.
> 1.2 As suggested, the motivation for adding a constant term is addressed in the revised paper, which is to relax the constraint of reaching Nash equilibrium such that it’s easier for the adversarial model to converge.
> 1.3 Yes, state ${s_t}$ is the hidden state vector. Sorry we did not explain clearly. It is clarified accordingly in the revised paper.
> 1.4 Yes, since a sentence is composed of $n$ words $\{{a_1},…{a_n}\}$, the reward of a sentence is the sum of the reward for each word, which is explained in section 5.2 in the revised paper .
> 1.5 Different from G-GAN in (Dai et al., 2017) that utilizes rollouts to learn the state value (expected future reward), which cannot give word-wise reward, our model learns the state action value ${f_{\theta,\phi}}$ that can be disentangled for each word under the form of rAIRL.
> In Eq. (11), although there’s expectation in the loss function, it disappears in the gradient by sampling the policy once using REINFORCE algorithm. And the gradient is averaged over mini-batches.
> In Eq. (5), the expectation is estimated by averaging over mini-batches for simplicity, following existing GAN paper in image captioning (Dai et al., 2017, Chen et al., 2019).
>
> 2
> 2.1 Regarding the compactness experiment, we agree that Figure 2 is not intuitive since it reflects compactness on the sentence level rather than on the word level. As suggested, the experiment of replacing a given word to compare the correlation between the reward differences and semantic differences is included accordingly in the revised paper. As shown in Table 1 in the revised paper, the reward difference of rAIRL correlates the best with the semantic difference for both similar words and distinct words, proving the compactness of the learned reward.
> 2.2 As suggested, a few visualized examples of the diversity are shown in Figure 4 in Appendix A in the revised paper.
> 2.3 Regarding the caption diagnosis experiment, we agree that utilizing the learned reward to diagnose and further improve the caption quality is a good idea. However, replacing only one word may not sufficient since the replaced word can affect the following words accordingly. For example, improving “a man is playing soccer” to “a man and a kid are playing soccer” needs to rewrite the sentence from “is”. Therefore, we chose to rewrite a sentence from the wrong word located by rAIRL. The full results are shown in Table 7 in Appendix B in the revised paper. The improvements on the scores compared to that of rewriting from a random position demonstrate that the proposed rAIRL can diagnose the caption at a word level, and further improves the caption quality by rewriting from the located position.

---

> > ### Comment · AnonReviewer1 · 2019-11-15
> > **Reply to the authors**
> >
> > I would like to say I appreciate the efforts that have been put in the response.
> > I'm happy to see a better study on the compactness is conducted in Table 1. Appendix B also includes empirical diagnostic results, as suggested.
> > Both of these studies significantly improve the value of this paper.
> > And the revised paper has addressed my concerns.
> >
> > Minor suggestions:
> > 1.  The diagnosis in Appendix B is very interesting, which could also include some qualitative samples before and after re-written.
> > 2.  I actually refer to the qualitative samples to show diversity as the diversity of captions generated for a single image, such as 5 diverse captions for an image. Of course, current samples in Figure 4 are also valuable.
> > 3. Since the deductions in section 4 are not straightforward, it will be good if deduction details are included in the appendix, which may make this paper more impactful.

---

> > > ### Author Response · Authors · 2019-11-15
> > > **Response to reviewer 1**
> > >
> > > We're grateful that you review our revision again and provide some further suggestions. We have updated the revision again according to your new comments. We have made the following updates:
> > >
> > > 1) We provide examples of the re-written captions in Appendix B.
> > > 2) We thought ''diversity'' referred to the diversity on a corpus level, meaning that similar pictures have notably different descriptions. And thus we provide Figure 4 in the last revision. According to your suggestion, we further provide examples of the top-5 generated captions of a single image in Figure 5. However, our model do not produce diverse top-$k$ descriptions because our aim is to learn a compact reward function in the discriminator. Driven by such reward function, the generator describes a given image with semantically similar words. Therefore, the top-5 captions in Figure 5 show compactness of the learned reward instead of diversity on a instance level.
> > > 3) The deduction details are presented in Appendix G.

---

### Official Review · AnonReviewer2 · 2019-10-23
**Official Blind Review #2**

**Rating:** 1

**Review:**

The authors propose using a recent method for adversarial inverse reinforcement learning (AIRL) for the task for generating high-quality image captions. Leveraging the GAN framework, a discriminator is trained to distinguish real captions from those produced by the generator, while the generator is optimized with policy-gradients (REINFORCE) to maximize the pseudo-reward from the discriminator. The main difference from prior work seems to be that the discriminator acts on a word-level, rather than sentence-level (as done, for instance in Dai et. al. 2017). Correspondingly, the generator policy is updated with the objective of 1-step reward maximization (more like contextual bandits), rather than with a long-term sequential decision-making objective (as done in Dai et. al. 2017). The evaluation is done using 2 data splits – standard and robust, with various metrics such as SPICE, CIDEr, BLEU, CHAIR. Diversity analysis and ablations are also performed to dissect the performance of the proposed approach.

My 2 main issues with the paper are confusing motivation (in section 1) and various imprecise parts (in section 3 and 4).

1.	The authors argue that current GAN-based captioning models provide ill-defined rewards due to the “reward ambiguity problem”. This problem is not explained or motivated well in the paper, but instead the readers are referred to the AIRL paper. “Reward ambiguity” in inverse-RL arises because there could be many reward functions that yield the same optimal policy. The AIRL algorithm recovers one of these possible reward functions, and since such a recovered reward could be shaped by the environment dynamics, AIRL attempts to disentangle the reward from the dynamics. The motivation there is to use the recovered reward on a new system with different transition dynamics. In the context of this paper though, I would like to understand the angle of reward ambiguity. The authors disentangle the sentence reward into word-wise rewards; however, I’m not sure if there’s any relation between this and the disentanglement done in AIRL for solving reward ambiguity.

2.	One of the objectives is learning compact rewards. It is claimed that addition of a constant term to the reward provided to the generator policy results in this, but what’s the intuition behind this? As for evaluation, it needs to be shown that words with similar semantics have similar discriminator score. How do we conclude this from Figure 2.? Also, please include Up-Down method results in Figure 2.

3.	Section 3 questions
     a.	How is state s_t defined? It is very hard to follow sections 3 and 4 without a clear definition and example for this.
     b.	“Finn et. al. 2016 proved that IRL is mathematically equivalent …” --- this is imprecise. Maximum-Entropy-IRL is equivalent to the GAN formulation, not general IRL.
     c.	p_theta(a,s) is referred to as “reward distribution”. I don’t think it’s a distribution.
     d.	Equation 3. AIRL defines g only as a function of state (s_t) for the disentanglement, and not like what the authors have written.
     e.	Equation 4. How is s_t sampled?

4.	Section 4 questions
     a.	4.1 says discriminator “maximizes the divergence”. This doesn’t seem correct.
     b.	f is referred to as state-value. This doesn’t seem correct.
     c.	Shouldn’t the -1 term in Equation 8 disappear under expectation?
     d.	Don’t understand how second line of Equation 11 is arrived at.

There are quite a few other sources of mathematically imprecise writing that I noticed. I would recommend the authors to be more robust in their presentation.


**Experience Assessment:**

I have published one or two papers in this area.

**Review Assessment: Checking Correctness Of Derivations And Theory:**

I assessed the sensibility of the derivations and theory.

**Review Assessment: Checking Correctness Of Experiments:**

I assessed the sensibility of the experiments.

**Review Assessment: Thoroughness In Paper Reading:**

I read the paper thoroughly.

---

> ### Author Response · Authors · 2019-11-12
> **Response to Review #2**
>
> Thank you for your valuable suggestions that could help us to improve our paper. But we are confused about the rating, which is very different from that of the other senior reviewers.
>
> We have tried our best to improve our paper and uploaded the revised draft. Below please find our response to your suggestions and questions. The motivation of our paper is to disentangle word-wise reward from different image-caption pairs, which is analogous to disentangling action reward form different system dynamics in AIRL. According to your detailed comments, we provide our explanations and revisions below.
>
> 1) Regarding the reward ambiguity problem, it can be explained from two perspectives: (1) from the perspective on the sentence level in image captioning, learning reward for a whole sentence is ambiguous since which word(s) causes the reward to increase or decrease is not accounted for, and thus there are many optimal policies that determine the sentence can explain one reward. An example is that each image in MS COCO has five ground truth captions. Although these captions may vary in formats, each caption has the same reward value in GAN; (2) from the perspective on the system level in AIRL, learning reward from different image-caption pairs is analogous to learning reward form different system dynamics in AIRL, which makes the discriminator unable to distinguish the true reward from those shaped by the system dynamics.
>
> The second point above is addressed by citing the AIRL paper because we think it’s intuitive to draw an analogy between different image-caption pairs in image captioning and different system dynamics in the AIRL paper. To clarify the point for readers less related to image captioning, the second point is further explained in section 1 in the revised paper.
>
> 2) Regarding the compact reward, as is stated in the fourth paragraph, learning compact rewards is achieved through AIRL by disentangling word-level reward. Adding a constant term does not lead to compact reward directly but facilitate it by refining the AIRL algorithm.
>
> Regarding Figure 2, we agree that it’s not straightforward to show compactness on the sentence level. As suggested by Review #1, a more intuitive experiment showing word-level compactness is included in the revised paper, where a given word is replaced to show the correlation between the reward differences and semantic differences. As is shown in Table 1 in the revised paper, the reward difference of rAIRL correlates the best with the semantic difference for both similar words and distinct words, proving the compactness of the learned reward.
> Regarding the Up-Down method, note that the Up-Down method using RL loss has been included in Figure 2, which is referred to as RL. The Up-Down method using MLE loss is not presented since Figure 2 demonstrates reward compactness only for reward-driven methods.
>
> 3)
> a. In the context of image captioning, state ${s_t}$ refers to the hidden state vector, which can be understood as the feature representation of the previous ${t-1}$ words. This is explained accordingly in the revised paper.
> b. ‘IRL’ will be corrected to ‘Maximum-Entropy-IRL’.
> c. Technically speaking, ${p_{\theta}}$ is indeed not a distribution without the normalization term sum($p_{\theta}$). It will be corrected to ‘The goal is to estimate $p_{\theta}$ …’.
> d.  This seems to be a misunderstanding. Eq (3) in our paper is exactly Eq. (4) in the original AIRL paper (Fu et al., 2018), which is a function of both action {a} and state {s_t}: $f_{\theta,\phi}(s,a,{s^\prime}) = g_{\theta}(s,a) + γh_{\phi}({s^\prime})−h_{\phi}(s)$.
> e. Since ${s_t}$ is the hidden state, it is not sampled but computed by sampling the previous words ${a_1,...,a_{t-1}}$. And each word is sampled according to the policy $\pi_{\psi}$.
>
> 4)
> a. This seems to be a misunderstanding. “maximize the divergence” is correct since the sum of the expectation represents the f-divergence, which is minimized by the generator and maximized by the discriminator (Roth et al., 2017).
> b. It is corrected to ‘the reward function $f$’ as in the AIRL paper (Fu et al., 2018).
> c. This seems to be a misunderstanding. Firstly, in REINFORCE algorithm, the expectation disappears first by sampling the policy once to estimate the expectation itself (step 2 to step 3 in Eq. (8)). Secondly, the -1 term is generated by taking a derivative over $\pi_{\psi}$ (step 3 to step 4 in Eq. (8) results in $-{\frac{1}{{\pi}_{\psi}}}*(-1))$.
> d. The deductions are included in Appendix G in the revised paper. ${L_t}$ is arrived at simply by taking the integral of the derivative $\nabla{L_t}$, and $\nabla{L_t}$ is computed exactly the same as Eq. (8).
>  As suggested, the mathematically imprecise writing has been improved in the revised draft.

---

### Official Review · AnonReviewer3 · 2019-10-23
**Official Blind Review #3**

**Rating:** 6

**Review:**

This paper proposes a refined Adversarial Inverse Reinforcement Learning (rAIRL) to remedy the reward ambiguity by decoupling the reward for each word in a sentence, while the existing methods that utilize reinforcement learning to optimize evaluation score handle only sentence-level rewards. Furthermore, a conditional term is introduced in the loss function to avoid mode collapse and to increase the diversity of the generated captions. Throughout experiments on MS COCO show that the proposed method achieves state-of-the-art performance with several evaluation scores.

I think this is a good paper. The idea to disentangle the sentence-level reward into word-level ones with Adversarial Inverse Reinforcement Learning (AIRL) is highly motivated. Although the original AIRL has a problem that the convergence is slow, the authors introduce a constant term to shift on of the stationary points. As reported, this refinement surprisingly improves the performance and achieves state-of-the-art performance, while the original AIRL degraded the performance.

Although I lean to accept this paper, I have two comments.
- I would like to recommend that the source code to reproduce the result should be released.
- As discussed by the authors, the introduced constant term can be regarded as a baseline in REINFORCE. In (Rennie et al., 2017), a self-critical sequence is introduced as a baseline in REINFORCE. Therefore, it is notable that this paper proposes another type of baseline. I would like the authors to compare Att2in (Rennie et al., 2017) to the proposed method, and to discuss why rAIRL outperforms Att2in as shown in Table 5.

**Experience Assessment:**

I have published in this field for several years.

**Review Assessment: Checking Correctness Of Derivations And Theory:**

I assessed the sensibility of the derivations and theory.

**Review Assessment: Checking Correctness Of Experiments:**

I assessed the sensibility of the experiments.

**Review Assessment: Thoroughness In Paper Reading:**

I read the paper thoroughly.

---

> ### Author Response · Authors · 2019-11-12
> **Response to Review #3**
>
> Thank you for your valuable comments, we have updated and uploaded our paper according to the comments from all the reviews.  Below please find our response to your suggestions and questions.
>
> 1) Regarding the source code, due to the reason of double-blind peer review, it is temporarily available at https://1drv.ms/u/s!Ar9lQiJpkYDIwys06qTohmC2OWMK?e=vAYR5C, and will be released on GitHub in December.
>
> 2) Thank you for your suggestion, the self-critical method in (Rennie et al., 2017) is referred to as RL in our paper, and has been compared in Section 5.3. In (Rennie et al., 2017), Att2in denotes the model structure revised from (Xu et al., 2015). Since our method is applicable to existing model structures, the comparison results using Att2in structure have been shown in Table 7 in Appendix C.
>
> The self-critical method in (Rennie et al., 2017) is a REINFORCE based algorithm and is denoted as RL in our paper following (Li et al., 2019a). Comparison with the self-critical method (RL) has been fully addressed in Section 5.3 and shown in Tables 2,3,4 and 5. In conclusion, the proposed rAIRL outperforms the self-critical method (RL) because the disentangled reward from rAIRL is learned to be compact, whereas the handcrafted reward in the self-critical method (RL) is predefined for the whole sentence, and thus causes reward hacking. Sorry for the misunderstanding, we explain in section 5.1 in the revised paper that RL is the self-critical method for better clarity.

---

### Decision · Program_Chairs · 2019-12-19

**Decision:**

Reject

**Comment:**

The paper proposed a refined AIRL method to deal with the reward ambiguity problem in image captioning, wherein the main idea is to refine the loss function in word level instead in sentence level, and introduce a conditional term in the loss function to mitigate mode collapse problem.  The results show the proposed method improves the performance and achieves state-of-the-art performance.  However there are concerns from the reviewers that the motivation of the work was not well explained and some inprecise parts exist in the paper.  The concept of "reward ambiguity problem" is not properly addressed according the opinion of reviewer2.  I would like to see these concerns be well addressed before the paper can be accepted.